# Detecting Out-of-Distribution via an Unsupervised Uncertainty Estimation for Prostate Cancer Diagnosis

**Jingya Liu**[1,3]                                                                JLIU1@CCNY.CUNY.EDU
**Bin Lou**[1]                                                          BIN.LOU@SIEMENS-HEALTHINEERS.COM
**Mamadou Diallo**[1]                                      MAMADOU-DIALLO@SIEMENS-HEALTHINEERS.COM
**Tongbai Meng**[1]                                          TONGBAI.MENG@SIEMENS-HEALTHINEERS.COM
**Heinrich von Busch**[2]                              HEINRICH.VON_BUSCH@SIEMENS-HEALTHINEERS.COM
**Robert Grimm**[2]                                          ROBERTGRIMM@SIEMENS-HEALTHINEERS.COM
**Yingli Tian**[3]                                                                  YTIAN@CCNY.CUNY.EDU
**Dorin Comaniciu**[1]                                    DORIN.COMANICIU@SIEMENS-HEALTHINEERS.COM
**Ali Kamen**[1]                                               ALI.KAMEN@SIEMENS-HEALTHINEERS.COM
**ProstateAI Clinical Collaborators**[*]

[1] *Digital Technology and Innovation, Siemens Healthineers, Princeton, NJ, USA*
[2] *Diagnostic Imaging, Siemens Healthineers, Erlangen, Bavaria, Germany*
[3] *The City College of New York, New York, NY, USA*

## Abstract

Artificial intelligence-based prostate cancer (PCa) detection models have been widely explored to assist clinical diagnosis. However, these trained models may generate erroneous results specifically on datasets that are not within training distribution. In this paper, we propose an approach to tackle this so-called out-of-distribution (OOD) data problem. Specifically, we devise an end-to-end unsupervised framework to estimate uncertainty values for cases analyzed by a previously trained PCa detection model. Our PCa detection model takes the inputs of bpMRI scans and through our proposed approach we identify OOD cases that are likely to generate degraded performance due to the data distribution shifts. The proposed OOD framework consists of two parts. First, an autoencoder-based reconstruction network is proposed, which learns discrete latent representations of in-distribution data. Second, the uncertainty is computed using perceptual loss that measures the distance between original and reconstructed images in the feature space of a pre-trained PCa detection network. The effectiveness of the proposed framework is evaluated on seven independent data collections with a total of 1,432 cases. The performance of pre-trained PCa detection model is significantly improved by excluding cases with high uncertainty.

**Keywords:** Out-of-distribution Detection, Uncertainty Estimation, AutoEncoder, Prostate Cancer Diagnosis

## 1. Introduction

Prostate cancer (PCa) is the most common cancer and one of the leading causes of death by cancer in men (Sung et al., 2021). Early detection of PCa and proper intervention are essential to increase the survival rate. Multi–parametric and bi-parametric Magnetic Resonance Imaging (mpMRI and bpMRI) can help improve the early detection of PCa, avoid unnecessary biopsies and reduce the over-diagnosis of clinically insignificant disease (Ahmed et al.,

---

[*] A list of members and affiliations appears at the end of the paper

2017). Deep learning based Computer–Aided Diagnosis (DL-CAD) systems have shown significant advantages in assisting radiologists with PCa diagnosis (Winkel et al., 2021; Schelb et al., 2021). However, in previous studies most of the systems were only tested on the data collected from the same institution or acquired using very similar protocols. It has been shown that deep learning-based methods may yield overconfident and erroneous results on the samples with different distributions from the training data (Lakshminarayanan et al., 2017). In practice, very large variance can be observed in MRI data collected from different clinical institutions: the scans can be acquired from different vendors, using various coils and protocols, with different levels of artifacts, and reconstructed with diverse parameter settings. These discrepancies within collected MRI studies may cause the systematic differences in the output of deep learning models trained with specific set of cases with homogeneous settings (Yan et al., 2020). The lack of the ability to distinguish between the testing data with different distribution from training may lead to missed or false detection in actual clinical diagnosis. It is essential to learn the distribution of the training data, detect the out-of-distribution (OOD) samples during inference primarily to maintain the performance of a lesion detection model.

Recently, deep learning-based methods have brought significant benefits to automatically learn complex feature representations for OOD detection (Golan and El-Yaniv, 2018). Many efforts have been made to detect the abnormal regions in images or videos, such as lesion detection on X-ray (Wang et al., 2017) and disease screening on retinal images (Zhou et al., 2020). Most state-of-the-art approaches are based on image reconstruction via AutoEncoder (AE), which learns the distribution of normal data and predicts regions uncertainty by comparing the original images and reconstructed images (Zhou and Paffenroth, 2017). Since the model only requires understanding the common features of the in-distribution data, it has a significant advantage in the uncertainty detection of medical images with unknown outlier labels. However, there is a concern that the reconstruction model may have a high generalizability and may be able to reconstruct abnormal samples with relatively high quality as well, which results in missing outliers. To solve this, Gong *et al.* (Gong et al., 2019) and Park *et al.* (Park et al., 2020) proposed memory-augmented autoencoder methods, which employ a memory book to learn only the key common features of the training dataset to ensure that the reconstructed image is generated from the latent representations of in-distribution data. Usually, the reconstruction loss between the original and reconstructed images is assessed to detect the anomaly and optimize the autoencoder. Recently, (Shvetsova et al., 2021) proposed an alternative approach by introducing perceptual loss, which computes the difference between the features maps of those images using a VGG19 network pre-trained on ImageNet.

Most of the aforementioned approaches are designed to detect general anomaly patterns in specific regions of the images. Methods for detecting the outlier with respect to specific tasks have yet to be explored. For PCa diagnosis task, the general anomaly detection method might not be applicable because the prostate lesion itself can be identified as "anomaly" if many healthy gland images are present in the training set. One of the main challenges is the lack of ground truth labels for OOD samples. Scans metadata cannot be used to determine outliers because the current CAD system might be insensitive to some scanning factors but sensitive to the changes that are not easily visible to humans. Moreover, it is inappropriate to assign OOD labels based on the performance of the PCa

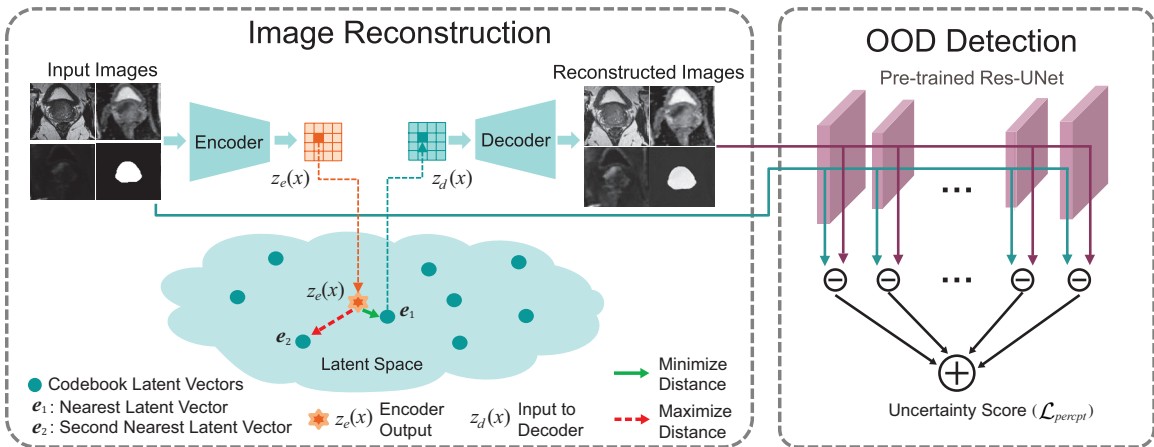

Figure 1: Schematic diagram of the proposed OOD detection framework. The framework consists of an image reconstruction network to learn the latent representation of in-distribution data and an OOD detection module to compute the uncertainty score based on the dissimilarities of the feature maps using a pre-trained PCa detection model (Res-UNet).

detection system, either. We do not expect the OOD detection model to be completely dependent on the PCa detection system output as it prevents the model from learning orthogonal features within the training data.

In this study, we propose an unsupervised OOD detection framework to estimate the uncertainty of an existing PCa detection model. The main contributions of this paper can be described in the following aspects: 1) The OOD samples are identified by comparing the similarity of the original and reconstructed images in the latent feature space defined by a pre-trained PCa detection model. This design makes the outliers to have higher uncertainty with respect to the PCa detection task. 2) The proposed uncertainty estimation model can be applied as an extension of an existing PCa detection system to provide the confidence of the detection on image data acquired under different conditions. 3) The performance of our proposed model is evaluated on a large testing dataset with 1,432 cases consisting of seven independent data collections acquired under a large variety of conditions. The performance of the pre-trained PCa detection model is significantly improved by excluding cases with high uncertainty scores, validating the hypothesis that identified OOD samples contributed to an average lower performance.

## 2. Methods

As shown in Figure 1, our OOD detection framework consists of an image reconstruction module and an uncertainty computation module. The first module learns embeddings of in-distribution data and makes the images reconstructed from the decoder similar to its original input. The second module assesses the uncertainty by comparing the similarity of the convolutional feature maps of a pre-trained (and fixed) PCa detection network between the original and reconstructed images.

### 2.1. Image Reconstruction Network

In order to force the model to focus on key features of the in-distribution data and lessen the generalization capacity of the autoencoder, we apply an embedding codebook to extract limited discrete latent representations and retain the high quality of reconstructed images. Different from the MemAE (Gong et al., 2019), which re-computes the latent vectors based on multiple similar latent vectors from the memory module, we propose to employ only the most relevant query in the memory book to gain higher sparsity in the latent space. This design resembles the Vector Quantized-Variational AutoEncoder (VQ-VAE) model (Oord et al., 2017; Razavi et al., 2019), which applies one-hot encoding to achieve quantization in the latent space. The reconstruction network is trained with in-distribution data to obtain the representations of normal samples from the input images $x$, and decode the extracted feature representations to generate reconstructed images $\hat{x}$. Instead of directly reconstructed from the encoded feature $z_e(x)$, the model first retrieves the most relevant item to $z_e(x)$ from the codebook (e.g. $e_1$ in Figure 1) and then takes it as the input $z_d(x)$ to the decoder. If the reconstruction network is trained with in-distribution data, the items in the codebook should capture representations of normal samples. The codebook contains $N$ different $K$-dimensional latent vectors, where $N$ determines the capacity of the memory and $K$ is the same as the dimension of bottleneck features.

During the training, to update the latent vector codebook, a gradient is computed to minimize the $L_2$ distance between the encoded feature $z_e(x)$ and its nearest embedding $e_1$, denoted as the compactness loss $\mathcal{L}_{compact}$ in Eq. 1. $sg(*)$ stands for a 'stop gradient' operation that has zero partial derivatives in the backpropagation, thus only optimizing embeddings in the codebook.

$$\mathcal{L}_{compact} = ||sg(z_e(x)) - e_1||_2. \tag{1}$$

Meanwhile, inspired by (Park et al., 2020), to increase the diversity of embeddings for learning various patterns of normal samples, a separateness loss $\mathcal{L}_{separate}$ is adopted to keep the distance of between $z_e(x)$ and its nearest embedding $e_1$ smaller than the distance between $z_e(x)$ and its second closest embedding $e_2$ with a margin $m$ (set to 1), as shown in Eq. 2.

$$\mathcal{L}_{separate} = [||sg(z_e(x)) - e_1||_2 - ||sg(z_e(x)) - e_2||_2 + m]_+. \tag{2}$$

### 2.2. Out-of-distribution (OOD) Detection

To evaluate the uncertainty relevant to the PCa detection, inspired by the deep perceptual autoencoder (Shvetsova et al., 2021), a perceptual loss is applied to optimize high-level structure learning specifically for feature-level representations of the in-distribution data. Unlike (Shvetsova et al., 2021) that utilized the VGG19 pre-trained on ImageNet, we select a previously validated PCa detection model to compare the feature map difference between original and reconstructed images. The PCa detection model contains multiple convolutional layers capable of extracting information at different scales. Since these extracted feature maps contain essential semantic and contextual information for PCa detection, we believe measuring dissimilarities in this feature space is more accurate than in the original space. The outliers identified in the feature space are more relevant to the specific task (i.e.,

PCa diagnosis) than general OOD samples. This design also solves the problem of lacking ground truth labels of OOD samples by associating the OOD detection with the specific PCa diagnosis task.

The PCa detection network follows the network architecture proposed in (Yu et al., 2020), which is a 2D Res-UNet with five residual blocks in the encoder and four blocks in the decoder. This Res-UNet serves as a fixed PCa detection model for training our OOD detection framework. The features maps of all nine residual blocks are used for computing the perceptual loss.

Denoting the $i^{th}$ residual block feature maps of the original image and the reconstructed image as $f_i(x)$ and $f_i(\hat{x})$ respectively, the perceptual loss is computed as the sum of $L_2$ distances between feature maps within the prostate region of all residual blocks: $\mathcal{L}_{percpt} = \sum_i ||f_i(x) \odot M(x) - f_i(\hat{x}) \odot M(x)||_2$, where $M(\cdot)$ is the binary mask of prostate and $\odot$ means element-wise multiplication. A higher perceptual loss indicates larger observed dissimilarities in the feature space and greater distance to the in-distribution data. This perceptual loss is used as the uncertainty score of the input image. The uncertainty scores of all 2D slices are averaged as the score of the case.

## 2.3. Objective Function

The total training objective function for the proposed OOD detection framework consists of compactness loss $\mathcal{L}_{compact}$, separateness loss $\mathcal{L}_{separate}$, reconstruction loss $\mathcal{L}_{rec}$, and perceptual loss $\mathcal{L}_{percpt}$, weighted by a set of hyper-parameters $\lambda_i$ (Eq. 3). The reconstruction loss is defined as the $L_2$ distance between decoder output and original input: $\mathcal{L}_{rec} = ||x - \hat{x}||_2$ for penalizing the intensity differences.

$$\mathcal{L} = \lambda_1 \mathcal{L}_{compact} + \lambda_2 \mathcal{L}_{separate} + \lambda_3 \mathcal{L}_{rec} + \lambda_4 \mathcal{L}_{percpt}. \tag{3}$$

## 3. Experimental Results and Discussions

### 3.1. Datasets and Evaluation

In this work, we used a large dataset of 2,746 cases (46,987 slices) collected from seven institutions, all of which have bpMRI prostate examinations and voxelwise annotations of the lesion boundaries. The ground truth labels are built based on radiology reports and carefully reviewed by an expert radiologist. 80% of the cases are used for training and 20% for validation. To obtain higher image quality of in-distribution samples, we have further evaluated the pre-trained PCa detection network on all the training data and excluded around 10% slices that have inaccurate predictions.

We also selected seven independent datasets with a total of 1,432 cases for testing. To better evaluate the OOD detection performance, we adopted OOD sample enrichment in the testing dataset by including cases acquired under a large variety of conditions. Three datasets ($\approx$ 53% cases) are significantly different from the training data. Many factors can impact the image quality and could result in high uncertainty with respect to PCa detection. More details of the testing datasets are described in Appendix A.

As there is no definite label of OOD samples, unlike the supervised problem, we could not directly compute the accuracy of outlier detection. To evaluate the efficacy of the proposed uncertainty estimation model, we compared the case-level PCa detection performance of

cases at different uncertainty levels. Our motivation is that OOD samples with higher uncertainty scores are cases that have higher likelihoods to be dissimilar to training samples used in the PCa detection model, and thus have worse performance than the in-distribution data. This is achieved by gradually rejecting cases with top-$k\%$ uncertainty score from the testing dataset and calculating the Area Under Curve (AUC) of the remaining cases using the pre-trained PCa detection model. The rejecting ratio is varied from 0% to 90% at the interval of 10%. 0% means all testing cases are included, which is the baseline performance without any uncertainty information.

### 3.2. Experimental Settings

The image reconstruction network takes 3-channel bpMRI images (T2-weighted, ADC, DWI b-2000) and binary prostate mask as input and outputs four corresponding reconstructed images. All input images are resized to $240 \times 240$. Detailed descriptions of the MRI preprocessing can be found in Appendix B.

A two-level hierarchical architecture (VQ-VAE2 (Razavi et al., 2019)) is utilized as the backbone network of the proposed reconstruction module, where the local and global information of in-distribution data are modeled separately. The latent representation at the bottom level simulates the detailed texture information, and geometric structures are encoded on the top level. First, the $240 \times 240$ input images are downsampled to $60 \times 60$ by the factor of 4 and then quantized as the bottom layer latent map. Second, the image is further downsampled to a size of $30 \times 30$ for quantizing the top layer latent feature. The decoder reconstructs the image based on the latent maps from both layers. Latent vectors with the size of $N = 256$ and $K = 512$ are applied in the codebook for quantization.

We conducted extensive experiments for hyperparameters tuning on the weights of different losses. Best performance was achieved using $\lambda_1$, $\lambda_2$, $\lambda_3$, and $\lambda_4$ values set to 0.1, 0.05, 0.15 and 0.7, respectively. During the training, the learning rate was set to $5 \times 10^{-4}$ and reduced by 0.1 after every 30 epochs. A total of 100 epochs was applied with a batch size of 32. We performed the evaluation on the model with the minimum validation loss. The weight decay was set to $3 \times 10^{-4}$, and the network was updated by Adam optimizer. During the inference, the perceptual loss was computed to evaluate the uncertainty.

### 3.3. Model Comparison

The proposed OOD detection model is applied on the entire testing dataset and case-level detection AUCs are computed at different rejecting ratios (Figure 2(A)). Our proposed model showed significant improvement on the detection performance of 6.3% after rejecting 60% of cases with high uncertainties. We also compared the AUC of all rejection levels with state-of-the-art methods AE, MemAE and VQ-VAE. Our framework outperforms other methods at most of the top-$k\%$ rejected percentiles. This result demonstrates benefits of applying discrete latent representations for lessening the capacity of the autoencoder.

Figure 2(B) further illustrates the uncertainty score distributions of each testing dataset. The proposed model assigns cases of Dataset $A - C$ with relatively high uncertainty scores and most of Dataset $D - P$ cases with low uncertainty scores. This result is consistent with the metadata information that datasets $A - C$ contain lots of cases acquired under significantly different conditions.

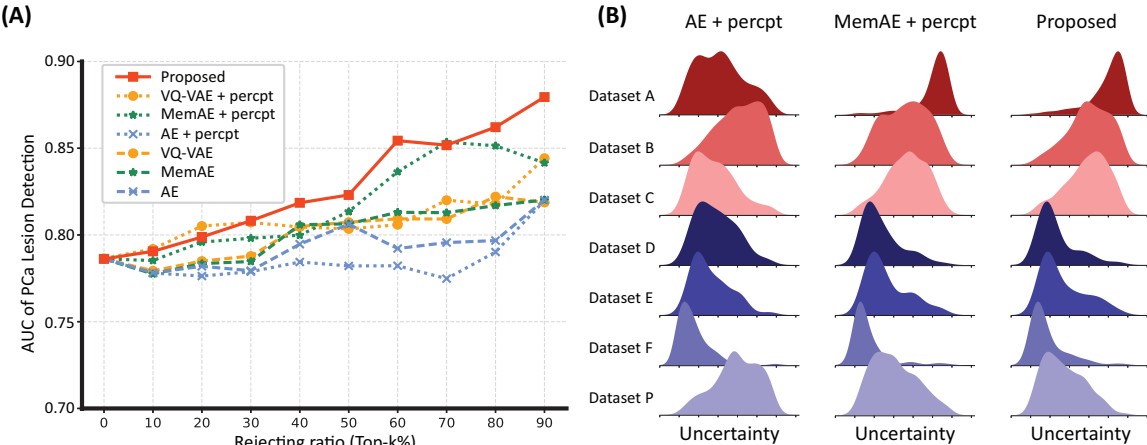

Figure 2: (A) Performance on the case-level AUC comparison with the state-of-the-art AE, MemAE, and VQ-VAE by rejecting top-$k$% uncertainty data from 10% to 90% with the interval of 10%. (B) Uncertainty score distributions for each testing dataset.

We also performed ablation studies to demonstrate the effectiveness of the proposed OOD detection model with results reported in Table 1. For this ablation study, the compactness, separateness, and reconstruction loss functions are excluded from the overall loss computation one at a time. We also compared the uncertainty score computation between using reconstruction loss and perceptual loss. As shown in Table 1, the OOD detection performance degrades if $\mathcal{L}_{percpt}$, $\mathcal{L}_{separate}$, or $\mathcal{L}_{rec}$ is excluded from training. Additionally, we observed computing uncertainty score using perceptual loss has significant advantages over the one computed using reconstruction loss. We believe assessing uncertainty based on perceptual loss has a stronger association with PCa detection task, mainly because evaluating distance in the features space is more contextual and is related to task at hand (i.e., PCa detection) as opposed to evaluating distance in the raw image space.

Table 1: Performance comparison between different ablation experiments and uncertainty score computation methods.

| Models | Uncert. | 0% | 10% | 20% | 30% | 40% | 50% | 60% | 70% | 80% | 90% |
|---|---|---|---|---|---|---|---|---|---|---|---|
| w/o $\mathcal{L}_{percpt}$ | $\mathcal{L}_{percpt}$ | 0.786 | 0.787 | 0.785 | 0.793 | 0.804 | 0.809 | 0.801 | 0.813 | 0.806 | 0.788 |
| w/o $\mathcal{L}_{rec}$ | $\mathcal{L}_{percpt}$ | 0.786 | 0.788 | 0.802 | 0.809 | 0.809 | 0.813 | 0.808 | 0.808 | 0.816 | 0.823 |
| w/o $\mathcal{L}_{separate}$ | $\mathcal{L}_{percpt}$ | 0.786 | 0.791 | **0.805** | 0.807 | 0.804 | 0.803 | 0.806 | 0.820 | 0.818 | 0.844 |
| **Full** | $\mathcal{L}_{percpt}$ | 0.786 | **0.791** | 0.799 | 0.808 | **0.819** | **0.823** | **0.854** | **0.852** | **0.862** | **0.880** |
| w/o $\mathcal{L}_{percpt}$ | $\mathcal{L}_{rec}$ | 0.786 | 0.780 | 0.785 | 0.789 | 0.806 | 0.807 | 0.810 | 0.812 | 0.824 | 0.815 |
| **Full** | $\mathcal{L}_{rec}$ | 0.786 | 0.780 | 0.788 | 0.796 | 0.805 | 0.817 | 0.814 | 0.821 | 0.836 | 0.831 |

### 3.4. Visualization

We also calculated the pixel-level uncertainty heatmap by averaging the feature map difference of all nine residual blocks of the pre-trained Res-UNet. All feature maps were resampled to the original input size of $240 \times 240$ to better align with the MRI input images. Since we only consider the perceptual loss within the prostate region, there is no uncertainty computed outside the gland. This result is displayed in the last column of Figure 3. The

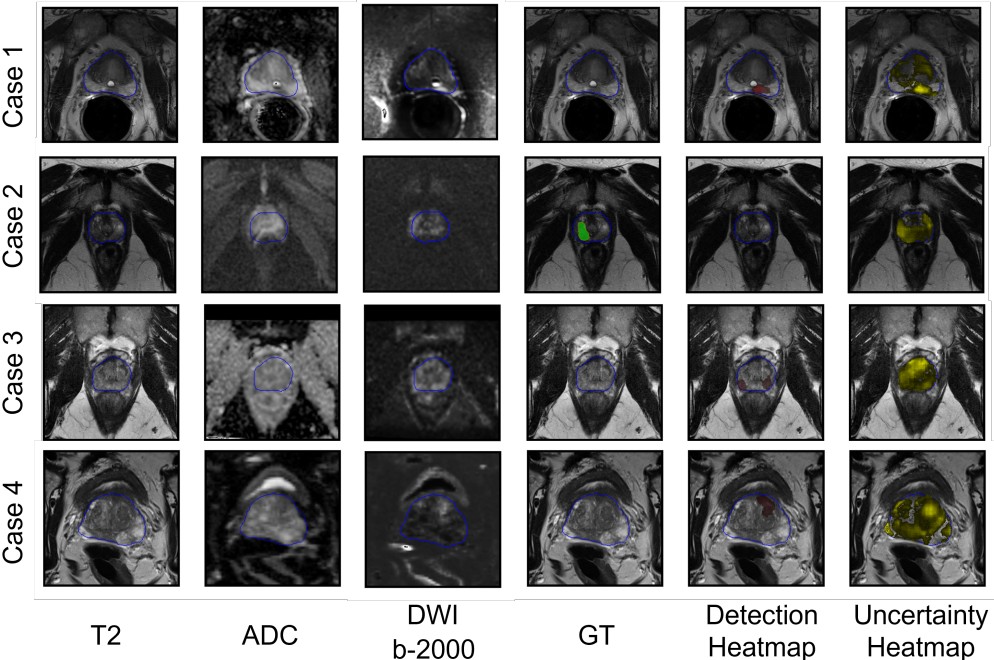

Figure 3: Visualization of four sample cases with high uncertainty scores.

T2-weighted, ADC, and DWI b-2000 images are shown in the first three columns. The prostate mask is shown as the blue contour on all images. The ground truth lesion annotation (marked as green), the lesion prediction heatmap (marked as red), and uncertainty heatmap (marked as yellow) are overlaid with the T2 images for better visualization.

Case 1 and case 4 show the high uncertainty on false-positive prediction regions. The hyperintensity artifact induced by the endorectal coil on the DWI b-2000 image is also highlighted in the uncertainty heatmap (Case 1). Case 2 is a false negative detection based on the pre-trained detection model, but the OOD detection model indicates high uncertainty in the lesion area. Case 3 shows high uncertainty on almost the entire prostate region. This is likely due to different DWI acquisition protocols used for this case as compared to cases within the training set.

## 4. Conclusion

This paper proposes an unsupervised OOD detection framework to learn the feature representations of normal samples using in-distribution data only and to predict the uncertainty score based on the semantic-level comparison between the original and reconstructed images. The uncertainty score can be applied as an additional score to prevent the overconfident and potentially erroneous PCa detection. This framework has the potential to be generalized into the OOD detection on different downstream tasks, such as classification. Our future work will focus on utilizing the learned latent representations of in-distribution data to update the pre-trained PCa detection model, which not only enables us to identify high uncertainty cases but also helps us to continually improve the overall detection performance.

**Disclaimer** The concepts and information presented in this paper are based on research results that are not commercially available. Future availability cannot be guaranteed.

**ProstateAI Clinical Collaborators** David Winkel[4], Henkjan Huisman[5], Angela Tong[6], Tobias Penzkofer[7], Ivan Shabunin[8], Moon Hyung Choi[9], Pengyi Xing[10], Dieter Szolar[11], Steven Shea[12], Fergus Coakley[13], Mukesh Harisinghani[14]

[4]Universitätsspital Basel, Basel, Switzerland. [5]Radboud University Medical Center, Nijmegen, NL. [6]New York University, New York City, NY, USA. [7]Charité, Universitätsmedizin Berlin, Berlin, Germany. [8]Patero Clinic, Moscow, Russia. [9]Eunpyeong St. Mary's Hospital, Catholic University of Korea, Seoul, Republic of Korea. [10]Radiology Department, Changhai Hospital of Shanghai, China. [11]Diagnostikum Graz Süd-West, Graz, Austriam. [12]Department of Radiology, Loyola University Medical Center, Maywood, IL, USA. [13]Diagnostic Radiology, School of Medicine, Oregon Health and Science University, Portland, OR, USA. [14]Massachusetts General Hospital, Boston, MA, USA.

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

## Appendix A. Testing Dataset Description

We selected 1,432 cases with bpMRI scans from multiple clinical institutions for the testing. Some datasets ($A$-$C$) were acquired under significantly different conditions from cases used for training. Therefore, cases from these datasets are likely to have higher uncertainties, when using the pre-trained model for PCa detection. Factors that might impact the data distribution of each dataset are listed in the following table.

Table 2: Testing dataset description

| Dataset | # Cases | Comment |
|---|---|---|
| $A$ | 244 | from independent clinical institutions, different scanner vendor from the training set, **with endorectal coil** |
| $B$ | 258 | from independent clinical institutions, different scanner vendor from the training set, **non-standard DWI protocol for prostate MRI** (e.g., low b-value 200 sec/mm$^2$, high b-value 2000 sec/mm$^2$) |
| $C$ | 266 | from independent clinical institutions, different scanner vendor from the training set, **full pelvis DWI sequence (low spatial resolution)** |
| $D$ | 115 | mixture of independent cases from two institutions, similar acquisition protocol as in the training set |
| $E$ | 105 | independent testing cases from the same institutions as the training set |
| $F$ | 101 | cases from independent clinical institutions, similar acquisition protocol as in the training set |
| $P$ | 343 | public ProstateX-Challenge dataset (Litjens et al., 2014), similar acquisition protocol as in the training set |

## Appendix B. Implementation Details

### MRI Data Preprocessing

We applied the same preprocessing pipeline as the one proposed in (Yu et al., 2020) and (Winkel et al., 2020). The pipeline first parsed and filtered the entire DICOM prostate study to load only the T2-Weighted (T2W) and diffusion weighted imaging (DWI) series. The apparent diffusion coefficient (ADC) maps were computed based on two DWI volumes with a low b-value of 0-100 sec/mm$^2$ and a high b-value of 800-1000 sec/mm$^2$, according to the suggestions in the Prostate Imaging Reporting and Data System (PI-RADS) guideline. We also adopted a logarithmic extrapolation to compute a new DWI volume with b-value at 0 and 2000 sec/mm$^2$ (DWI b-0 and b-2000). All DWI sequences were aligned to the T2W volume through rigid registration and then all re-sampled to the size of $240 \times 240 \times 30$ with a voxel spacing of 0.5mm $\times$ 0.5mm $\times$ 3mm.

All bpMRI images were normalized to facilitate training process. The T2W volumes were linearly normalized to [0, 1] based on the 0.05 and 99.95 percentiles of the volumes' intensities. The DWI ADC volumes were normalized by a constant value of 3000 as they are quantitative parametric maps. The DWI b-2000 volumes were first normalized by the median intensity within the prostate gland region of the corresponding DWI b-0 volumes, and then normalized by a constant value to map the range to [0, 1].

**PCa Detection Network**

We used the 2D Res-UNet proposed in (Yu et al., 2020) and (Winkel et al., 2020) as the pre-trained PCa detection network. The preprocessed T2W, DWI ADC, DWI b-2000 images and the binary prostate mask were concatenated as the input to the network. The network was trained with the same training and validation data of the OOD detection network. The binary cross entropy (BCE) loss and Dice loss were applied during the training. As routine clinical practice mainly uses patient-level analysis to recommend biopsy testing, in this study, we evaluated the PCa detection performance by computing patient-level AUC. Specifically, all slices in each case were assessed by PCa detection model to obtain a 3D heatmap. We defined the maximum value of the 3D heatmap as the prediction score of the patient. The ground truth labels were defined based on the radiology reports - cases with PI-RADS$\geq$ 3 lesions were considered as positive.