# OpenReview forum: "Detecting Out-of-Distribution via an Unsupervised Uncertainty Estimation for Prostate Cancer Diagnosis"
_MIDL.io/2022/Conference — MIDL 2022_

### Official Review · Reviewer_Xnyn · 2022-01-22

**Confidence:** 3
**Preliminary Rating:** 3
**Recommendation:** Poster

**Summary:**

The authors present an unsupervised framework to estimate the uncertainty of Prostate cancer detection models using out of distribution (OOD) detection. The proposed method solely relies on in-distribution data and thus does not require the labeling process of OOD data. Based on the similarity between the reconstructed and original images, the OOD samples are identified. The authors provide an evaluation of their proposed method in combination with an ablation study to determine the effectiveness of the OOD detection model.

**Strengths:**


- The formal definition of the different loss parts helps grasping the construction of the overall loss
- Motivation and methodology is well written and fairly understandable
- “Additionally, we observed computing uncertainty score using perceptual loss has significant advantages over the one computed using reconstruction loss.” [p. 7] → I appreciate how the authors backed this claim up with the Ablation Study (Table 1)
- Dataset Description in the Appendix is interesting and good but could be a little bit more specific


**Weaknesses:**

My main concern is how the results are presented. I described below my concern in two parts.
- “Our proposed model showed significant improvement on the detection performance of 6.3% after rejecting 60% of cases with high uncertainties.” [p. 5] → Kind of obvious, if you remove all the uncertainty, the network is pretty certain in its segmentation and the results are getting better, no ?
- Figure 2A: Being the best when rejecting most of the data is not really a good approach to show the method's strengths. Rather the performance in the lower rejection rates is very important and interesting. → Maybe the authors should make a figure where the differences can be seen for the 0 to 30% rejection rates. This would show the performance differences between the methods way better than it is now.


**Deanonymize Review:**

no

**Detailed Comments:**

- 3 cases are significantly different [p. 5]: What does this significantly mean and how is it determined ?
- Did the resizing of the images somehow influence the amount of artifacts in the scan so it could influence the uncertainty score and if so, how did the authors cope with it ? “All DWI sequences were aligned to the T2W volume through rigid registration and then all re-sampled to the size of 240×240×30 with a voxel spacing of 0.5mm × 0.5mm × 3mm.” [p. 12]
- Ablation Study: The authors should do the ablation study for every possible combination, since the order of removing the losses will have an impact on the performance after all → Alternatively the authors should at least mention this point and why they did use this specific order.

**Final Rating After The Rebuttal:**

3: Borderline

**Justification Of The Final Rating:**

It's good to see how the authors addressed my comments. It also shows that they have some research wrt to those comments.
I would suggest that the authors include the gist of their comment to the image resampling since they did not mention any of it in the paper. With this added information I think the paper would be out of the woods.

**Paper Type:**

both

**Questions To Address In The Rebuttal:**

Please answer my question about the presentation of the results. May be there are some misunderstanding from my part. In that case, I am happy to reconsider my overall rating.
But if that's not the case, please tell me why the authors presented such a lopsided result?

**Special Issue:**

no

---

### Official Review · Reviewer_Bq7a · 2022-01-24

**Confidence:** 5
**Preliminary Rating:** 4
**Recommendation:** Poster

**Summary:**

This paper propose an OOD detection framework as a means to improve the usability of the downstream task of detecting of prostate cancer. The OOD detector outputs uncertainty scores that are fine-tuned for a separate prostate cancer detection model. Samples with high uncertainty (i.e. are considered OOD) are then considered inappropriate/rejected for the detection model and thus excluded from the test data to improve classification performance.

**Strengths:**

- Combination of OOD and detection model, where OOD model provides information whether it is appropriate to apply the detection model in the first place.
- From a clinical perspective, this is great as blindly applying detection/diagnosis models is not optimal. Preceding the detection model with a gatekeeper model that judges the appropriateness of applying that detection model makes sense for clinical applications.
- Overall nice evaluation, showing how rejecting data, for which the detection model is inappropriate, helps to improve the applicability of the detection model

**Weaknesses:**

- the individual components are not really novel, the contributions listed in the intro section are overstated
- the use of a gatekeeper OOD model is not really novel either
- missing information on hyperparameters

**Deanonymize Review:**

no

**Detailed Comments:**

Major:
- The novelty of some of the stated contributions are questionable
    - Re contribution (1) : It is not the first to do unsupervised inlier learning (e.g. MemAE, f-AnoGAN)
    - Re contribution (2): This is the same as MemAE
    - Re contribution (3): I guess not using VGG is main novelty here
    - Main novelty is the use of an OOD model as a gate keeper for a detection model in a clinical application
- Not all hyperparameter values are given
    - What is K? (dimension of codebook)
    - What is m? (margin in $L_{separate}$)
- In Fig 2A, it would be nice to see AUC on 100% of the validation data  to get an upper bound on the model’s performance
- Validation data distribution in Figure 2B would be informative
- AUC does not take class imbalance into account
    - Maybe use FPR @95TPR
- Uncertainty scores are not normalized

**Minor**
- Misusing the word uncertainty, “outlier score” or “rejection score” would be better
- At the very least, appendix should have info about model and its hyperparameters
    - Is it exactly the same as VQ-VAE2? Any modifications? K? m?
- A validation case should also be visualized in Section 3.4
    - Ideally this shows very low uncertainty in the heatmap
- VQ-VAE2 citation in Section 3.2 is incorrect


**Final Rating After The Rebuttal:**

4: Weak Accept

**Justification Of The Final Rating:**

Thanks for the rebuttal, which helped clarify several of the issues raise in our review. This does not result in a change/raise of the rating (it was already at a weak accept), as the proposed novelty is still limited.

**Paper Type:**

both

**Questions To Address In The Rebuttal:**

- What is the detection score used by the PCa detection model?
- How would you determine a threshold for new samples at test time?
- Please add the requested hyperparameter information
- Address the major points addressed in the comments

**Special Issue:**

no

---

### Official Review · Reviewer_ctpp · 2022-01-24

**Confidence:** 5
**Preliminary Rating:** 4
**Recommendation:** Poster

**Summary:**

This paper proposes a method to derive uncertainty maps associated to supervised semantic segmentation based on a UNET architecture. The whole pipeline is based on two cascaded networks – one autoencoder that takes the grey level images as input and outputs its reconstruction and – a pre-trained UNET based model that takes as input the original or reconstructed grey level image and outputs the binary segmentation. This model is trained end-to-end based on a global loss that is a weighted sum of different loss terms including a contribution of the standard autoencoder loss as well as a perceptual loss term that measures the distance between original and reconstructed images in the feature space of the second U-Net segmentation network. This perceptual loss is used to derive uncertainty maps, that are used to detect out-of-the distribution samples. The application domain is that of Prostate cancer detection based on multi-parametric MRI. The effectiveness of the proposed framework is trained on a series of about 2400 patient datasets retrieved from 7 clinical centers and evaluated on seven independent test dataset encompassing 1432 data, some acquired in the same clinical acquisition setting as the training datasets, some in with different acquisition settings.

**Strengths:**

-Uncertainty estimation is a hot topic in the community in order to detect erroneous results (false negative or false positives) in the test set.
-Novelty : the estimation of the uncertainty maps based on conceptual loss term computed from the feature maps of derived from the original and reconstructed images.
-the proposed method is generic and can be applied to different tasks, eg classification
-The study is based on a large experimental cohort


**Weaknesses:**

-The objective of the paper is not clear. The authors propose a method to derive uncertainty maps, which is fair and well-conducted; The questionable part of the paper, however, concerns the strong assumption made by the authors that test samples with higher uncertainty scores are outliers of the in-distribution training data. It is not clear how results reported in this study allow validating this hypothesis. I thus suggest the authors either to clarify this part of the paper by rewording this section and adding other analyses to confirm this hypothesis or to focus on uncertainty estimation without relating their contribution to the detection of out-of-the distribution outliers.
-Detailed description of the experiments should be added (see below) to confirm the soundness of the experiments.


**Deanonymize Review:**

no

**Detailed Comments:**

-Please provide details of the PCa cancer detection task : Could you confirm that the pre-trained U-Net based network in Figure 1 outputs a segmentation of the PCa lesion? How is this network pre-trained (ie which loss, which training dataset)? The text mentions ‘case-level’ detection AUC, could you confirm that AUC is computed at the patient level? If so, how is the segmentation map outputted from the Res-UNet converted into a binary prediction (contains or do not contain lesions)?

-Please provide details of the comparison with state of the art with AE, MemAE and VQ-VAE

- The authors mention that uncertainty scores are computed at the slice level and averaged over all 2D slices of a case to derive a patient score. Regarding segmentation task, it might be more of interest to correlate the slice-based uncertainty score with dice metric at the slice level. Did the authors consider this idea?

- As mentioned in the previous section, the hypothesis relating cases with high uncertainty to OOD samples should be further explored.
1) The authors may consider reporting AUC performance for each dataset separately. If the authors’ hypothesis is verified, we expect lower performance for datasets A to C). It would also be nice to report as well the % of each dataset in the topk% rejected data. Same comments regarding datasets A to C should apply, the topk% rejected data should contain most of cases from these datasets.
2) Analysis of failure. It would also be interesting to analyse the most uncertain cases of the in-distribution test data (datasets D to G) to better understand why they were rejected. Hypothesis is that they were rejected because they are out of the distribution…This hypothesis should be confirmed based on the visual analysis of the uncertainty maps to better understand the spatial distribution of most uncertain areas (equally spread over the prostate?, located close to the border, inside lesions?..)


**Final Rating After The Rebuttal:**

4: Weak Accept

**Justification Of The Final Rating:**

I thank the authors for their thorough review and added methodological details in the revised version of the paper. I maintain my grading of ‘weak accept’, a deeper analysis of the outlier cases would help increasing the soundness of the study.

**Paper Type:**

both

**Questions To Address In The Rebuttal:**

The authors should address the elements reported in the two previous sections, especially related to the link between the uncertainty metrics derived by the authors and the detection of out-of-the distribution samples.

**Special Issue:**

no

---

### Meta-Review · Area_Chair_Zk55 · 2022-02-15

**Recommendation:** Accept (Poster)
**Confidence:** 4

**Metareview:**

With this work the authors make a step toward the clinical application of disease detection systems, here for prostate cancer. The system identifies out-of-distribution samples in an unsupervised way using an uncertainty measure obtained after training an autoencoder.
Being able to deal with data sets that might differ from the one used during training is very important. Even though the approach is not entirely innovative, its performance is well explored in the experiments that rely on large data sets. The authors made good use of the rebuttal phase as they manage to clarify several points raised by the reviewers. In future work, they could consider using synthetic data to confirm or not their hypotheses.

---

### Decision · Program_Chairs · 2022-02-28

Accept